# Using Femtosecond Laser Light to Investigate the Concentration- and Size-Dependent Nonlinear Optical Properties of Laser-Ablated CuO Quantum Dots

**DOI:** 10.3390/nano14201674

**Published:** 2024-10-18

**Authors:** Mohamed Ashour, Rasha Ibrahim, Yasmin Abd El-Salam, Fatma Abdel Samad, Alaa Mahmoud, Tarek Mohamed

**Affiliations:** 1Laser Institute for Research and Applications LIRA, Beni-Suef University, Beni-Suef 62511, Egypt; 2High Institute of Optics Technology HIOT, Sheraton Heliopolis, Cairo 11799, Egypt; 3Department of Engineering, Faculty of Advanced Technology and Multidiscipline, Universitas Airlangga, Surabaya 60115, Indonesia

**Keywords:** nonlinear optics, femtosecond laser, high repetition rate, copper nanoparticles, quantum dots, nonlinear absorption, nonlinear refraction, optical limiter

## Abstract

In this work, the nonlinear optical (NLO) properties of CuO nanoparticles (CuO NPs) were studied experimentally using the pulsed laser ablation (PLA) technique. A nanosecond Nd: YAG laser was employed as the ablation excitation source to create CuO NPs in distilled water. Various CuO NPs samples were prepared at ablation periods of 20, 30, and 40 min. Utilizing HR-TEM, the structure of the synthesized CuO NPs samples was verified. In addition, a UV–VIS spectrophotometer was used to investigate the linear features of the samples. The Z-scan technique was utilized to explore the NLO properties of CuO NPs samples, including the nonlinear absorption coefficient (β) and nonlinear refractive index ($n_{2}$). An experimental study on the NLO features was conducted at a variety of excitation wavelengths (750–850 nm), average excitation powers (0.8–1.2 W), and CuO NPs sample concentrations and sizes. The reverse saturable absorption (RSA) behavior of all CuO NPs samples differed with the excitation wavelength and average excitation power. In addition, the CuO NPs samples demonstrated excellent optical limiters at various excitation wavelengths, with limitations dependent on the size and concentration of CuO NPs.

## 1. Introduction

While investigating the various features of nanomaterials, scientists and researchers have been overwhelmed. The noticeable and unexpected change in the properties of materials of these very small sizes, in contrast to their larger counterparts’ qualities, has fascinated researchers. This has had a significant impact on the development of numerous applications that are in use today as well as plans to create new applications based on the general features of these materials [1,2]. The credit goes to those whose curiosity has been aroused and who are interested in studying the properties of nanomaterials and the change in their properties caused by changing their shape and size. The presence of these materials has enriched many life applications, such as biological, chemical, engineering, and medical applications [3,4,5,6].

Among the most-often manufactured nanoparticles are Au, Ag, Pt, Cu, Pd, Re, Zn, Ru, Co, Cd, Al, Ni, and Fe. Owing to their enhanced optical, optoelectrical, catalytic, antibacterial, antiviral, and cancer-fighting properties [7,8,9,10,11,12], metal nanoparticles are very intriguing materials for a wide range of practical applications [13].

When it comes to optoelectronic [14], biomedical [15], gas sensing [16], and nonlinear optical applications [13,17], copper oxide nanoparticles (CuO NPs) are the most valuable among the metal nanoparticles. CuO NPs have garnered significant interest from enterprises across multiple industries because of their exceptional surface plasmon resonance (SPR) in the visible spectrum a few years ago. In optics, the surface absorption plasmon of CuO NPs can express distinct colors depending on the particle shape, size, and condensation rate. Furthermore, electricity can be used to make high-temperature superconducting materials because the superconductivity transition temperature rises to the point where the particle diameter is small (less than 1 nm) [18]. With the rapid development of several methods for producing CuO NPs with a wide range of sizes and shapes, several new possibilities for nonlinear optical applications have become possible.

There are three basic techniques for NPs synthesis: physical, chemical, and biological. The physical technique is also known as the top-down approach, whereas the chemical and biological approaches are commonly referred to as bottom-up approaches [19]. The biological approach is also known as the green system of NPs. Top-down techniques separate bulk materials into nanostructured materials. Top-down techniques include mechanical milling, laser ablation, etching, sputtering, and electro-explosion [20]. The bottom-up method, also known as the constructive method, includes the construction of materials ranging from atoms to clusters to nanoparticles. Bottom-up processes include chemical vapor deposition, sol-gel, spinning, pyrolysis, and biological synthesis [21].

Pulsed laser ablation is a very promising technique for creating nanoparticles (NPs) since it is stable, low cost, and very effective. Furthermore, the laser ablation technique may be used to create nanoparticles with a variety of properties, exhibiting the amazing nonlinear optical (NLO) properties of NPs, depending on the experimental laser settings [22,23].

While several studies have investigated the NLO characteristics of CuO NPs, no thorough studies utilizing femtosecond laser pulses have been carried out at various concentrations and sizes of CuO NPs or with various laser parameters (e.g., excitation wavelength and power). For example, the authors of [13,17,23,24,25,26] studied the NLO characteristics of CuO NPs for a certain excitation wavelength and CuO NPs size utilizing a variety of production techniques.

In the present work, the NLO of a variety of CuO NPs samples was studied. PLA was utilized to create the CuO NPs samples at various ablation times (20, 30, and 40 min) via a Nd: YAG nanosecond laser. The morphology, size, and shape of the NPs samples were investigated using TEM, and their concentration was assessed via the ICP technique. A UV–VIS spectrophotometer was used to measure the linear optical characteristics. The nonlinear optical properties (nonlinear refractive index $n_{2}$ and nonlinear absorption coefficient β) of all the samples were measured using the Z-scan technique. Furthermore, the produced CuO NP samples were considered useful for the application of the optical limiter based on the findings of nonlinear optical measurements.

## 2. Synthesis and Structural Characterization of CuO NPs

### 2.1. Synthesis of CuO NPs by PLA

The pulsed laser ablation approach is thought to be the most effective, dependable, and economical method for synthesizing CuO NPs [22,27,28,29,30], in addition to being the best in terms of nanoparticle size distribution. The experimental setup employed in this investigation to synthesize CuO NPs is depicted in Figure 1. To produce the CuO NPs, Spectra Physics (Milpitas, CA, USA) supplied a Quanta-Ray PRO-Series 350-10 nanosecond 532-nm Nd: YAG laser source with a 10 Hz repetition rate, and a pulse duration of 10 ns was used. A square piece of bulk Cu sample $\left(25\;\mathrm{mm}\;\times \;20\;\mathrm{mm}\;\times \;2\;\mathrm{mm}\right)$ with a purity of 99% was exposed to a pulsed laser beam while it was submerged in a glass beaker with 10 milliliters of pure distilled water to form CuO NPs colloids, where the average excitation power was fixed at 600 mW.

As shown in Figure 1, three mirrors were used to direct the pulsed laser beam, which was then focused onto the bulk Cu sample using a 10 cm convex lens. To prevent water from splashing, which might decrease the quality of the ablation process, the beaker was fixed on a rotary stirrer at a uniform and controlled speed. When a powerful laser beam impacts a metal surface over an extended length of time, it leaves deep cavities. This produces oscillations in the laser fluence, which influences the target and may reduce the efficiency of CuO NPs synthesis. To avoid this, a motorized sample spinner moves the target continuously throughout the ablation process. A rotatable platform rotates the beaker carrying the target around a central axis, allowing the laser to reach different locations of the target (Figure 1). In addition, a spinner rotates the beaker containing the target and water to homogenize the solution and removes the nanoparticles from the laser path, preventing them from absorbing the incident laser beam and reducing ablation efficiency. The laser ablation times (LATs) of 20, 30, and 40 min were sufficient to produce CuO NPs colloids that were stable and free of aggregation.

In the nanosecond range, laser ablation is a more complicated event that involves numerous interconnected processes both during and after the laser pulse ends. Heat transfer; laser and intrinsic plasma radiation transfer; the creation and propagation of shock waves; contact boundaries in the vaporized matter and the surrounding gas environment; heterogeneous and homogeneous phase transformations occurring in the evaporated substance; the formation and expansion of a plasma plume, etc. are examples of these processes [31].

Nanosecond-laser ablation can be explained mathematically by a hydrodynamic multiphase model [31,32] that takes into consideration plume generation, target heating, the removal of mass mechanisms, and plasma generation. In this model, one could presume that the laser energy gathered by the electrons is instantly redistributed and transmitted to the lattice when modeling ns-laser ablation. Since the target’s lattice and electron subsystem reached thermal equilibrium, target heating may be explained by the conventional heat conduction formula. The target will move with a specific recession velocity on its surface as a result of the material being eliminated (ablated) [32,33].

### 2.2. Characterization of CuO NPs

A high-resolution transmission electron microscope (HR-TEM, JEM-2100, Joel, Tokyo, Japan) operated at 200 KV was used to investigate the average size and shape of the created CuO NPs. The TEM images of colloidal CuO NPs formed by laser light are shown in the insets of Figure 2a–c, and the micrographs show that the CuO NPs are spherical-like. The micrographs were analyzed using ImageJ software (origin 2018) to determine the average particle size. Figure 2a–c illustrates the histograms of the size distribution of CuO NPs at a constant average power of 600 mW and different ablation times of 20 min, 30 min, and 40 min. Gaussian fits of the histogram provided average nanoparticle sizes of 6.2 nm, 6 nm, and 5.6 nm, respectively. It can also be concluded that at a higher LAT, the average diameter of the CuO NPs becomes smaller, which is due to the fragmentation process [34].

The volume-specific surface area (VSSA) method [35,36] can be used to compute the total surface area (SA) of the acquired samples using Equation (1), keeping in mind that the Cu density is $\rho =8.96\;g/{\mathrm{cm}}^{3}\;$(at room temperature). The extracted results are plotted in Figure 3, which shows that the total surface area increases with increasing LATs. This can be understood as the total number of CuO NPs increasing with increasing ablation time due to the fragmentation process. Additionally, the number of particles (*N*) at each gram is increased by increasing the LAT.
(1)SA=N·sa
where $N=\frac{V}{v}$ is the total number of particles in 1 g (where the total volume of the NPs (V=m/ρ)), the volume of one particle is $v=4\pi r^{3}$), and the surface area of one particle is $sa=4\pi r^{2}$. This is why nanoparticles exhibit unique properties due to their high surface area-to-volume ratio; as the SA-to-volume ratio increases, the fraction of atoms at the surface increases, and surface forces become more prominent [36,37].

An inductively coupled plasma-optical emission spectrometer (ICP–OES) was used to measure the CuO NPs’ molar concentration. Moreover, one can find out that the results in Figure 3 are correlated with the results in Figure 2, where the concentration is related to the total number of synthesized nanoparticles. While the ICP–OES technique was used, each series of measurement intensity calibration curves was constructed from a blank and three or more standards from Merck Company (Darmstadt, Germany).

The accuracy and precision of the metal ion measurements were confirmed via external reference standards from Merck, and standard reference materials and quality control samples from the National Institute of Standards and Technology (NIST) were used to confirm the instrument reading [38].

Additionally, as shown in Figure 4, the concentration of CuO NPs increased from 14 to 26 mg/L, as determined via ICP–OES; in the meantime, the volume fraction of CuO NPs can be computed using the following equations [39,40].
(2)$$V=\frac{V_{P}}{V_{P}+V_{L}}$$
where $V_{P}$ and $V_{L}$ represent the volume of distilled water and the volume of CuO NPs, respectively, according to Equation (1). As the particle mass disappears in distilled water, some simplifications and modifications to Equation (2) yield
(3)$$V=\frac{C_{Particles}}{C_{Particles}+\rho }$$
where $C_{Particles}$ represents the particle concentration of the CuO NPs obtained from the ICP–OES.

## 3. Linear and Nonlinear Optical Characteristics of CuO NPs

### 3.1. Linear Optical Characteristics of CuO NPs

The huge number of electrons in a material fluctuates when the electric field changes (similar to a photon). These oscillations are called plasmons with resonance frequency and are quantized [41,42]. The size, shape, and nature of the nanoparticles affect the plasmon resonance frequency (PRF). Planck’s constant is related to the plasmon energy (E=hv).

The interaction between incoming photons and surface plasmon resonance (SPR) dominates the optical characteristics of metal nanoparticles [41,43]. Plasmon frequencies in the visible range are exhibited by metal nanoparticles such as silver, gold, and copper [44,45,46]. This means that when polychromatic light (such as white light) impinges on the nanoparticle, the frequency that is related to the SPR will be absorbed. The size, shape, and spectrum position of a particular particle influence its surface plasmon resonance frequency and intensity [42,46].

The absorption and transmission spectra of CuO NPs samples were measured using a UV–VIS spectrophotometer (Peak Instruments C-7200, Inchinnan, United Kingdom) and are displayed in Figure 5, where the CuO NPs sample has certain absorption peaks. These peak patterns are all in the visible region and are caused by localized surface plasmon resonance (LSPR) [47]. The interband transition is the cause of another peak that emerges in the near-UV region [48,49].

From the obtained results shown in Figure 5, the energy bandgap of the CuO NPs can be estimated via Tauc’s plot method [50,51] and is shown in Figure 6.
(4)$${\left(\alpha hv\right)}^{2}=A\left(hv-E_{g}\right)$$
where α is the linear absorption coefficient, A is an energy-independent constant, and hv is the photon energy. Figure 6 shows that increasing the laser ablation time from 20 min to 40 min leads to a decrease in the energy bandgap from 3.38 eV to 3.11 eV. The physics behind this behavior can be understood from Equation (5), the well-known confinement energy equation, which is called the Brus equation [52,53,54]. Because of quantum confinement, the outermost radius of the quantum dot influences the wavelength of the light that is emitted. This equation shows how altering the radius of the nanoparticles influences the wavelength *λ* of the light that is emitted and, consequently, the energy band gap $E_{g}$ [53,54].
(5)$$\Delta E=E_{g}+\frac{h^{2}}{8r^{2}}\left(\frac{1}{m_{e}}+\frac{1}{m_{h}}\right)$$
where h is Blanck’s constant and r represents the radius of the nanoparticle, the mass of the excited electron $m_{e}$ and the excited hole $m_{h}$.

### 3.2. Nonlinear Optical Characteristics of CuO NPs

Numerous optical applications, including optical signal processing, optical computers, ultrafast switches, ultrashort pulsed lasers, sensors, laser amplifiers, and many more, rely heavily on nonlinear optics (NLOs) [55,56,57,58]. The NLO characteristics of materials may be measured via various methods, including nonlinear interferometry, Z-scans, and degenerate four-wave mixing [59,60,61]. The most appropriate method for this investigation was the Z-scan approach, which is thought to be an overly sensitive single-beam technique that measures the sign and magnitude of nonlinear optical coefficients via the principle of spatial beam distortion [62,63].

In this study, both the nonlinear refractive index ($n_{2}$) and nonlinear absorption coefficient (β) of the CuO NPs were measured using the Z-scan technique, where a closed aperture (CA) and an open aperture (OA) were employed simultaneously. Figure 7 shows the Z-scan setup where a femtosecond pulsed laser was used as an excitation light source.

The powerful, controllable pulsed femtosecond laser Inspire HF100 was used to irradiate the CuO NPs samples during the experiment. This beam’s spatial profile has an M2 < 1.1 Gaussian distribution with a TEM_00_ spatial mode. This type of femtosecond laser system had a 100 fs pulse duration, an 80 MHz repetition rate, and a wavelength tunable range of 345–2500 nm. A MAI-TAI HP mode-locked femtosecond Ti: Sapphire laser with an 80 MHz repetition rate and a wavelength range of 690 nm to 1040 nm was used to pump the Inspire HF100 laser. The Inspire HF100 and Mai-Tai HP femtosecond laser systems are manufactured by Spectra Physics, Inc., USA.

To direct the laser beam toward the CuO NPs sample, it was first passed through an attenuator (A) to regulate the laser’s incidence power, followed by a convex lens with a focal length of 5 cm. The CuO NPs sample was loaded into a micro quartz cuvette and moved ±2.5 cm using a translation stage across the lens focus in the z-direction. The quartz cuvette has a 1 mm optical length, a capacity of 350 µL, and dimensions of 52 mm × 1.5 mm × 3.5 mm. The nonlinear absorption coefficient β and the nonlinear refractive index $n_{2}$ were measured concurrently using a 50/50 beam splitter.

Since the entire transmission was monitored using a power meter (PM2) in the OA approach, minor distortions were rendered inconsequential, and fluctuations in the signal were solely the result of nonlinear absorption. To ensure that any nonlinear phase changes in the beam were exclusively caused by variances in the samples, the CA-approach concentrated beam went through a closed aperture in front of the power meter (PM1). The Newport 843 R power-meter model was the same in both the OA and CA configurations. Since 0.1<S<0.5 is the linear normalized transmittance value (S), this value was set to S=1 in the OA measurement and S=0.3 in the CA measurement. Notably, the sample thickness (L) satisfies the necessary condition $L\geq z_{o}/\Delta \Phi$, where $z_{o}$ is the Rayleigh range [64]. The experimental error in the obtained NLO parameters was about 10%, which mainly originated from the determination of the irradiance distribution used in the experiment, i.e., beam waist, pulse width, and laser power calibration.

#### 3.2.1. Measuring the Nonlinear Absorption (NLA) Coefficient

Using the OA configuration and irradiation, the CuO NPs samples were irradiated with a pulsed femtosecond laser, and the three CuO NPs samples exhibited positive NLA coefficients, with reverse saturable absorption (RSA) behavior, as illustrated in Figure 8a–c. Equation (6) provides the normalized transmittance expression [63].
(6)$$\Delta T_{OA}=1\pm \frac{\beta I_{0}L_{eff}}{{\left(m+1\right)}^{3/2}\left[1+{\left(z/z_{o}\right)}^{2}\right]}$$
where *z* is the sample’s relative location; $z_{o}$ is the Rayleigh length, which was calculated via $z_{o}=\pi {w_{0}}^{2}/n\lambda$ (where $w_{o}$ is the laser beam waist (16.8 µm ± 1.7 µm)); $I_{0}$ is the laser peak intensity at focus; and $L_{eff}$ is the effective length of the sample given by $L_{eff}=\left(1-e^{-n\alpha_{0}L}\right)/n\alpha_{0}$, where m=1 for two-photon absorption (2PA) and m=2 for three-photon absorption (3PA). Moreover, 2PA was determined to have the greatest fit in this investigation and to be consistent with the experimental findings.

The dependence of the NLA coefficient on both the average excitation power and the excitation wavelength was examined during this study, where the different CuO NPs samples demonstrated a reliance on both the excitation wavelength and the average power. Figure 8a–c shows the experimental results of the open-aperture (OA) measurements for the three different CuO NPs samples. During these measurements, the samples were irradiated to different average powers of 0.8 W, 1 W, and 1.2 W at a constant excitation wavelength of 800 nm.

Figure 9a,b illustrates the relationships between the NLA coefficient and both the average size of CuO NPs and the CuO NPs’ concentration. The effect of changing the average power can be concluded from Figure 9, where the NLA coefficient increases with increasing average power due to the small size of the CuO NPs at higher concentrations and thus the high surface-to-volume ratio.

Figure 10a–c shows the results of the OA measurements for the three different CuO NPs samples, where the samples were irradiated with different excitation wavelengths ranging from 750 nm to 800 nm at a constant average power of 1 W. All the CuO NPs samples exhibited RSA behavior that increased with increasing wavelength. Figure 11a,b shows how changing the excitation wavelength affects the NLA coefficient, taking into consideration both the average size and the concentration of the CuO NPs samples.

By raising the excitation wavelength and/or the average power of the input laser beam, the CuO NPs sample behaves in a way that guarantees that the excited-state absorption cross-section at a certain excitation wavelength *λ*, σ_ex_(*λ*), becomes larger than the ground-state absorption cross-section, σ_gr_ (*λ*), which is due to free carrier absorption (FCA) or multiphoton absorption [65,66].

Two-photon absorption (2PA) is a nonlinear optical process since it depends on the simultaneous absorption of two photons; its likelihood is related to the photon dosage (D), which is proportional to the square of the light intensity ($D\;\propto \;I^{2}$). It is good that by knowing the value of the NLA coefficient, the value of the 2PA cross-section (2PACS) can be determined via Equation (7) [35]:(7)$$2PACS=\frac{\beta }{N_{A}d_{o}10^{-3}}{\left(\frac{hc}{\lambda }\right)}^{2}$$
where $d_{o}$ (mol/L) is the concentration of CuO NPs, $N_{A}$ is Avogadro’s number, and hc/λ is the photon energy. Table 1 shows the variation in the 2PACSs as a function of both the CuO NPs concentration and incident average power. According to the extracted results, the data in Table 1 clearly match the behavior of the NLA coefficient, where the sample concentration has the greatest impact on the 2PACS.

#### 3.2.2. Measuring the Nonlinear Refractive Index (NLR)

Using the closed-aperture (CA) schema, different samples of CuO NPs were irradiated to different average powers at constant excitation wavelengths. In this setup, as illustrated in Section 3.2, the laser beam is sent through a closed aperture that is placed before the power meter (PM1), ensuring that any nonlinear phase shift is due to the sample translation at the z-axis around the focal length of the used lens, as shown in Figure 7. The variation in the percentage of the light transmitted through the samples is due to the Kerr lens effect [67,68].

The experimental results of these measurements are shown in Figure 12a–c, where all samples show defocusing performance, which means that the phase shift $\Delta \Phi_{0}$ has a negative sign as the peak precedes the valley. In this study, the CA Z-scan configuration was used to measure the mentioned samples at different average powers varying between 0.8 W and 1.5 W and pulsed at a constant excitation wavelength of 800 nm.

The femtosecond laser energy absorbed by the sample and is converted to heat, which leads to the induction of thermal heating during exposure. More heat builds up as a result of the sample not returning to its initial equilibrium temperature during the intervals between laser pulses. The temperature variation that results from this heat can alter the refractive index distribution. The cumulative thermal heating caused by a variation in the refractive index modifies the divergence and convergence of the subsequent incoming pulses, leading to an inaccurate interpretation of the CA Z-scan readings.

With w being the laser beam diameter and D being the thermal diffusion coefficient ($D=k/\rho C_{p}$), which depends on the sample parameters (k is the thermal conductivity, ρ is the sample density, and $C_{p}$ is the specific heat capacity), the resulting laser pulse separation time of 12.5 ns (for 80 MHz) was less than the sample’s thermal characteristic time ($t_{c}=w/4D$). The predicted thermal characteristic time for liquids and certain optical glasses exceeds 40 µs [69]. A mathematical technique that was employed in [35,70] was used in this study to address the issue of accumulative thermal lensing.
(8)$$\frac{1}{f\left(z\right)}=\frac{aLE_{P}{F_{l}}^{3/2}}{\omega {\left(z\right)}^{2}}\left(1-\frac{1}{\sqrt{N_{P}}}\right)$$

The fitting parameter is denoted by $a=\alpha \left(dn/dT\right)/2k\;{\left(\pi^{3}D\right)}^{1/2}$, while the temperature derivative of the refractive index is represented by dn/dT. *L* is the sample thickness, $\omega \left(z\right)$ is the radius of the laser beam at the sample, $E_{P}$ is the energy per laser pulse, $F_{l}$ is the repetition rate, and the total number of laser pulses impinging on the sample is $N_{p}=t\times F_{l}$ throughout the scan, and t stands for the exposure duration of each CuO NP sample, which is approximately 2 min in this experiment. The results of Equation (9) are used to calculate the NL refractive index of the different samples via Equation (8) [70,71].
(9)$$n_{2}=\frac{\lambda {w_{o}}^{2}\Delta \Phi }{2P_{peak}L_{eff}}$$
where ${w_{o}}^{2}$ is the beam waist, $P_{peak}$ is the peak power at different excitation wavelengths ($P_{peak}=\frac{P_{avg}}{t_{P}\cdot F_{l}}$, where $t_{P}$ is the pulse duration), and $L_{eff}$ is the effective length of the sample $L_{eff}=\left(1-e^{-2\alpha L}\right)/\;2\alpha$, where α is the linear absorption coefficient and where λ is the excitation wavelength. Δϕ is the phase shift expressed by:(10)$$\Delta \Phi =\frac{Z_{0}}{2f\left(0\right)}$$

Thus, with the sample at the focus, $f\left(0\right)$ is the focal length of the produced thermal lens.

Figure 13 shows the variation in the NL refractive index $n_{2}$ (and thus $\Delta T_{p-v}$) as a function of the average laser power for different CuO NPs samples. The relationship between both $n_{2}$ and the average power is linear. For each sample, the nonlinear refractive index parameter ($n_{2}$) was calculated based on the slope of the fitting lines. This can be understood as follows: when the average power increases, more electrons exit and enhance the nonlinear response of the sample.

In this experiment, the NL refractive index is measured on the basis of self-defocusing via a Kerr lens [72], where the sample acts as a negative lens. The Kerr effect causes an optical phase delay greater on the beam axis and less outside the axis when a short optical pulse propagates through a nonlinear medium. This is comparable to how a lens works—the wavefronts are distorted, defocusing the pulse. Using the predicted NL refractive index values $n_{2}$, the dioptric power $f^{-1}$ of the Kerr lens can be estimated via Equation (11) [73,74].
(11)$$f^{-1}=\frac{8n_{2}d}{\pi w^{4}}P_{avg}$$
where d is the sample thickness (in this case, it is the cuvette length) and w is the laser beam diameter. Figure 14 shows the variation in the dioptric power of the Kerr lens induced by the different CuO NPs samples as a function of the incident average power. The power of the induced lens clearly increases by increasing the magnitude of the average power, taking into consideration that the sample exhibits a negative Kerr lens.

This dioptric power of the Kerr lens is correlated with the CA measurements, where the nonlinear refractive index is directly proportional to the dioptric power of the induced lens.

## 4. Optical Limiter

To prevent high-irradiance light from transmitting while permitting low-irradiance light to do so, passive optical limiting takes advantage of a material’s nonlinear response; this function is comparable to that of photochromic sunglasses [75]. The main purpose of optical limiting is to prevent high-intensity input light from damaging delicate optical components [75,76]. Moreover, the optical limiter can be used to shield human eyes and other sensors from powerful laser radiation [77,78].

As an optical limiter (OL), the three synthesized samples at various LATs were evaluated. The samples were examined at various wavelengths between 750 and 850 nm, with the sample placed at the convex lens’s focal length and the input average power being scanned between 0.2 and 1.2 W. The samples exhibit the features of an efficient optical limiter. As the size of the CuO NPs decreased and the concentration of CuO NPs increased, as shown in Figure 15a–c, the sample behaved better as an optical limiter. This finding is consistent with the RSA values stated in Section 3.2.1, which reveal that samples with higher concentrations have greater NLA coefficient.

An ideal optical limiter exhibits a linear transmission below the threshold and a constant output intensity above it [79,80]. The CuO NPs sample functions as an ideal optical limiter as the surface area-to-volume ratio rises and as the concentration of CuO NPs falls, as may be seen from Figure 15a–c. To our knowledge, not much research has explored CuO NPs’ behavior as an OL. H. S. Sumantha and B. L. Suresha investigated CuO NPs with a spherical flake shape in [80] to ascertain the optical limiter’s reaction. A limiting threshold of 2.11 kW/cm^2^ was discovered. GS Boltaev et al. investigated the optical limiting impact of a copper oxide nanoellipsoid in [25], where the OL curve was observed at an energy of about 0.6 μJ for 800 nm with a 60 fs pulse laser source.

As far as we are aware, a few previous studies [14,17,25,77] measured the nonlinear optical parameters of CuO NPs samples. Nevertheless, no comprehensive research has examined the effects of adjusting the average power and excitation wavelength of CuO NPs samples prepared by the laser ablation method at different ablation times.

## 5. Conclusions

Using a Nd: YAG nanosecond laser, the laser ablation methodology was applied to synthesize CuO NPs in distilled water. The CuO NPs were created at various laser exposure times, a constant excitation wavelength of 532 nm, and an incident average power of 600 mW. The main features of the specimens were investigated using a UV–VIS spectrophotometer, TEM, and ICP–OES. As a function of the laser ablation time frame, which ranges from 20 to 40 min, respectively, a drop in the average size of CuO NPs was observed from 6.2 nm to 5.6 nm, respectively. As a result, the energy band gap reduced from 3.38 to 3.11 eV and the concentration of CuO NPs increased from 14 to 26 mg/L.

Moreover, a comprehensive study to explore the nonlinear optical properties (nonlinear absorption coefficient and nonlinear refractive index) of CuO NPs was conducted via the Z-scan technique using a femtosecond laser as an excitation source. The CuO NPs samples were exposed to different excitation wavelengths ranging from 750 to 850 nm and different average powers ranging from 0.8 to 1 W. The CuO NPs samples exhibit RSA and act as a defocusing material with a negative noncolinear refractive index. The nonlinear absorption coefficient and nonlinear refractive index of the different CuO NPs samples were found to be dependent on the laser excitation wavelength and average power.

Finally, the CuO NPs samples were investigated experimentally as an optical limiter at various excitation wavelengths ranging from 750 to 850 nm. It was observed that the samples exhibit good limiting at high incident power. Moreover, the limiting behavior of the CuO NPs samples was found to be dependent on NP size.

## Figures and Tables

**Figure 1 nanomaterials-14-01674-f001:**
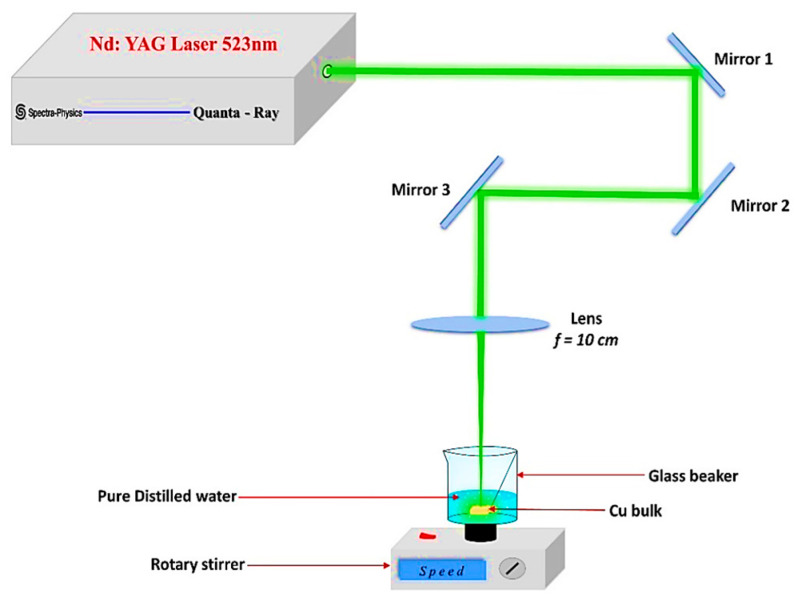
Schematic preview of the pulsed laser ablation process for synthesizing CuO NPs via a nanosecond Nd: YAG pulsed laser.

**Figure 2 nanomaterials-14-01674-f002:**
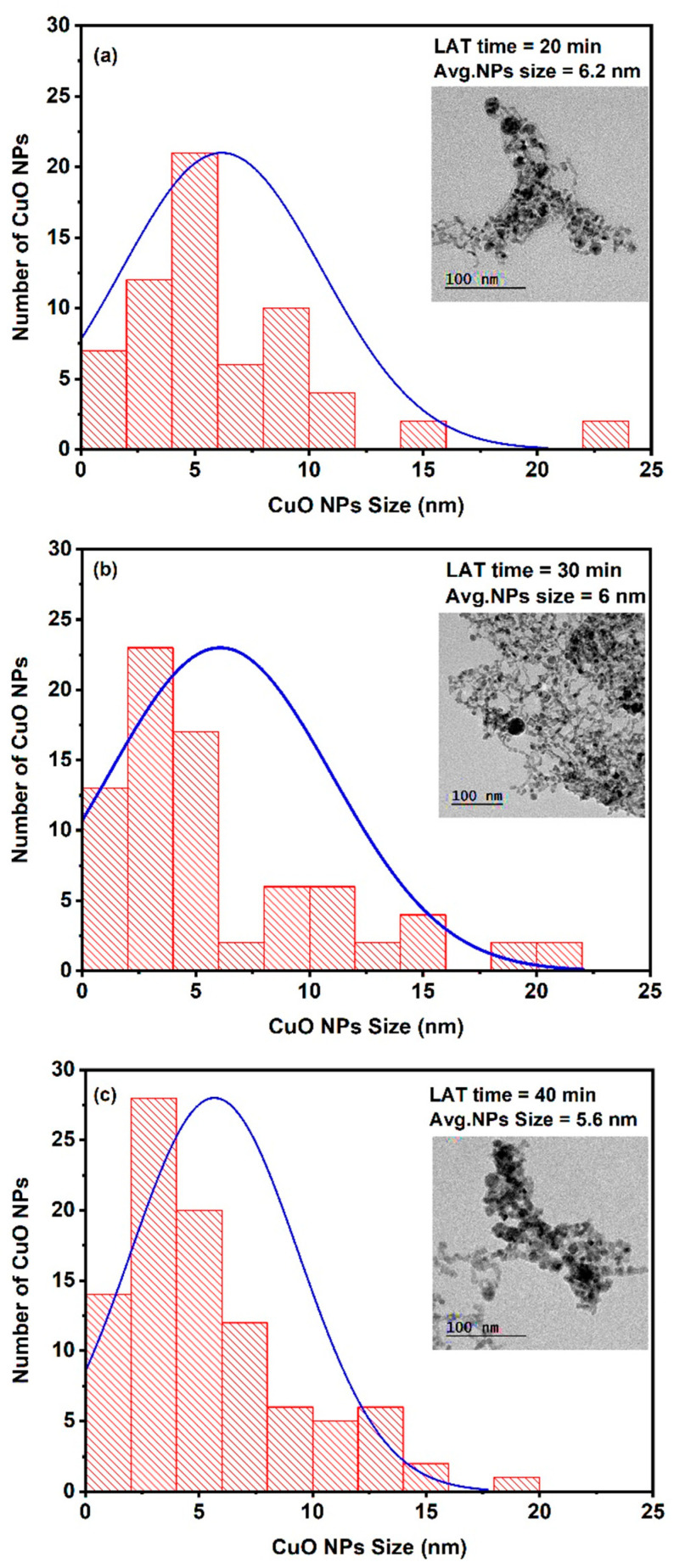
The size distribution of the CuO NPs ablated by the nanosecond Nd: YAG Laser (**a**) LAT = 20 min, (**b**) = 30 min, and (**c**) LAT = 40 min.

**Figure 3 nanomaterials-14-01674-f003:**
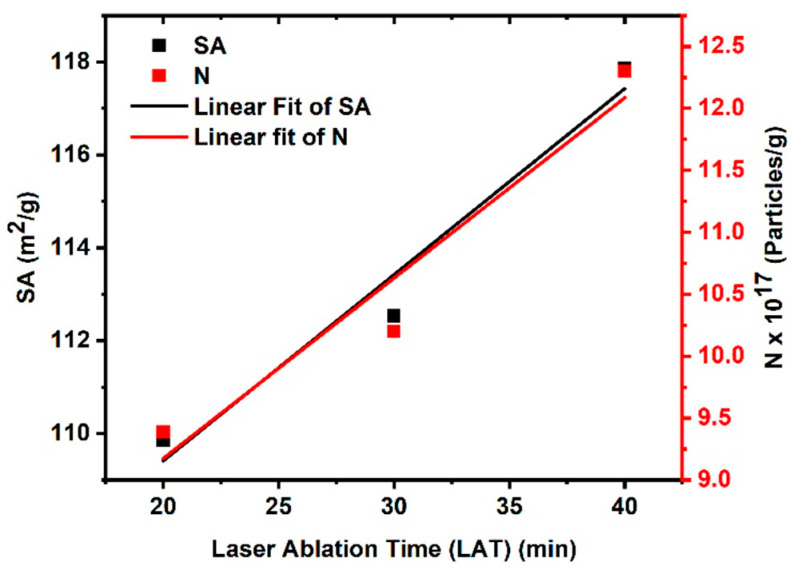
The laser ablation time (LAT) of the CuO NPs is a function of both the total surface area (left x-axis) and the number of particles per gram (right x-axis).

**Figure 4 nanomaterials-14-01674-f004:**
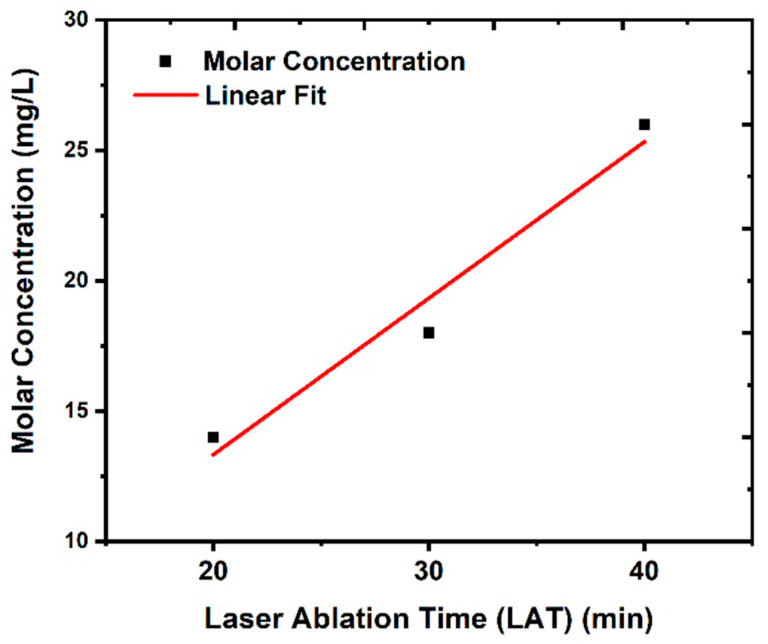
Molar concentration of CuO NPs determined via ICP–OES at different laser ablation times (LATs).

**Figure 5 nanomaterials-14-01674-f005:**
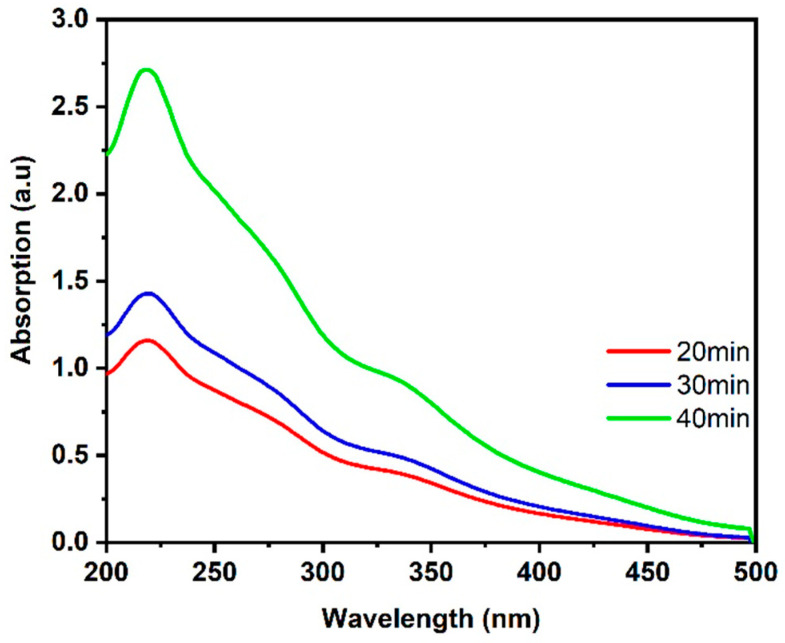
The linear absorption spectra of the three CuO NPs samples at LAT = 20, 30 and 40 min.

**Figure 6 nanomaterials-14-01674-f006:**
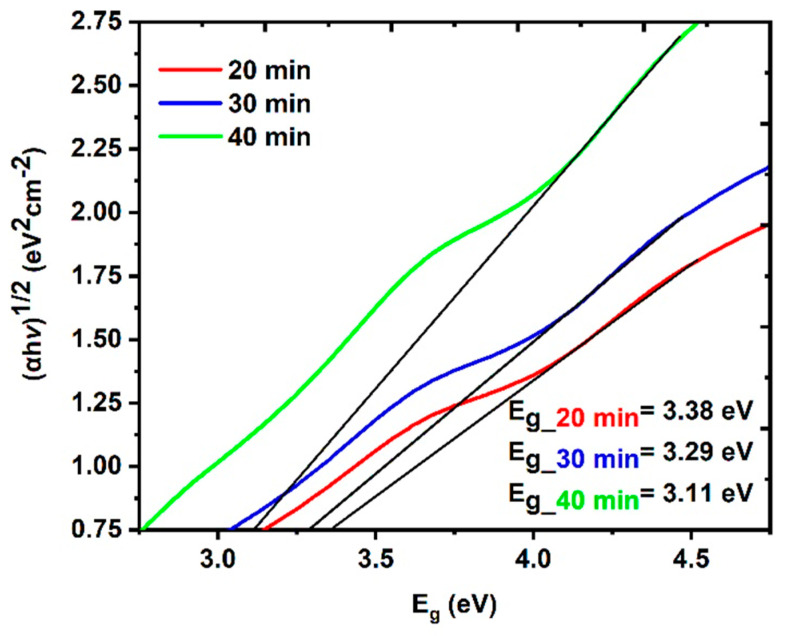
Energy band gap of the CuO NPs samples at different LATs.

**Figure 7 nanomaterials-14-01674-f007:**
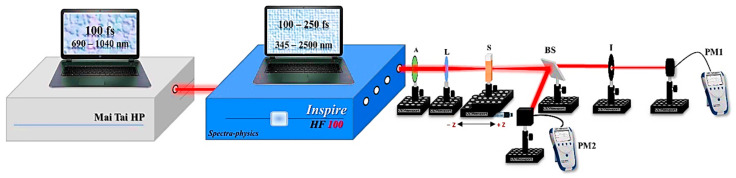
The experimental setup of the Z-scans: A, attenuator; L, convex lens; S, CuO NPs sample; BS, beam splitter; I, iris; PM, power meter.

**Figure 8 nanomaterials-14-01674-f008:**
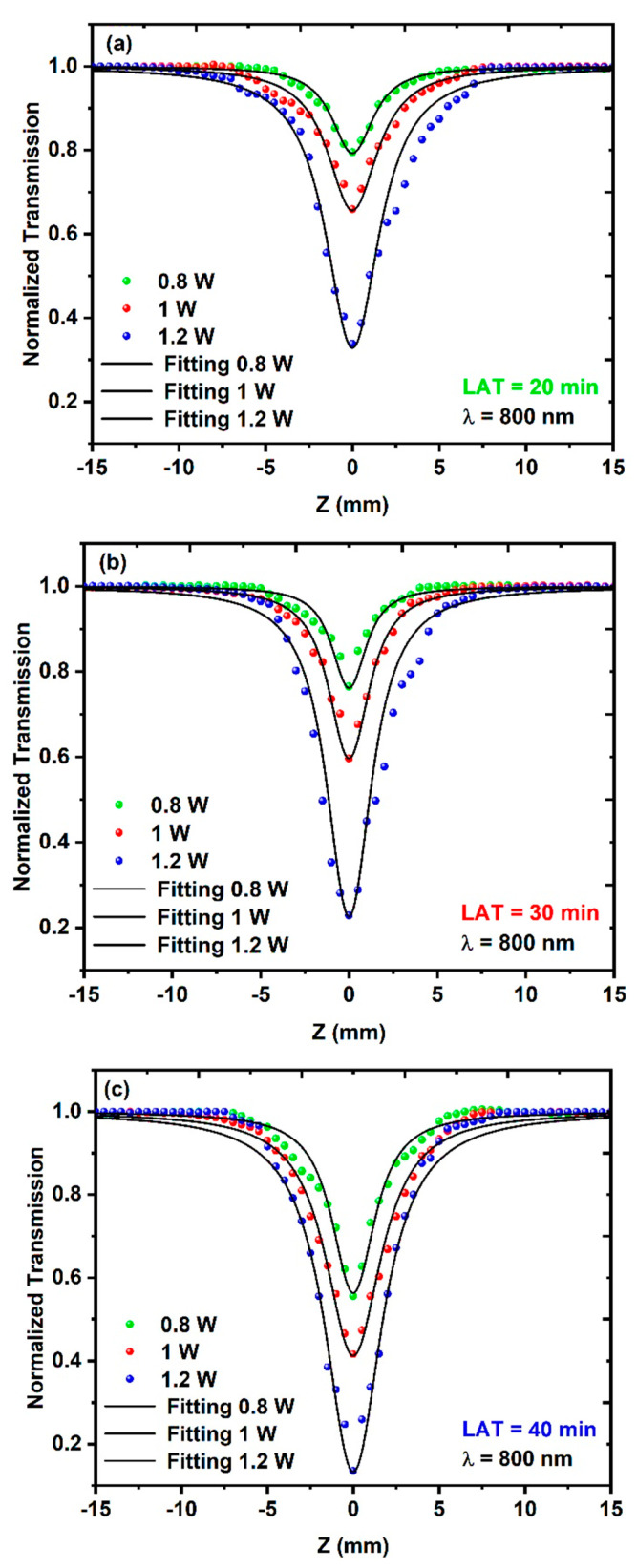
Open-aperture (OA) Z-scan measurements for different CuO NPs samples at a constant excitation wavelength of 800 nm: (**a**) LAT = 20 min, (**b**) LAT = 30 min and (**c**) LAT = 40 min.

**Figure 9 nanomaterials-14-01674-f009:**
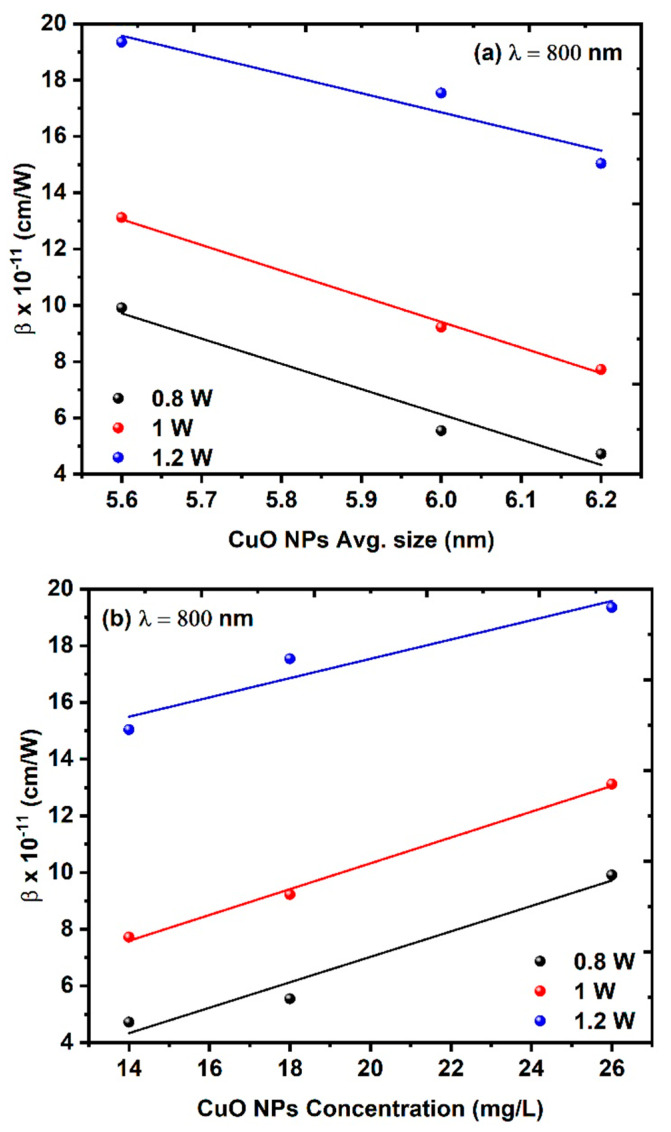
Effect of average power $P_{avg}$ on the NLA coefficient at a constant excitation wavelength of 800 nm: (**a**) NLA coefficient as a function of average CuO NPs average size and (**b**) NLA coefficient as a function of the CuO NPs concentration.

**Figure 10 nanomaterials-14-01674-f010:**
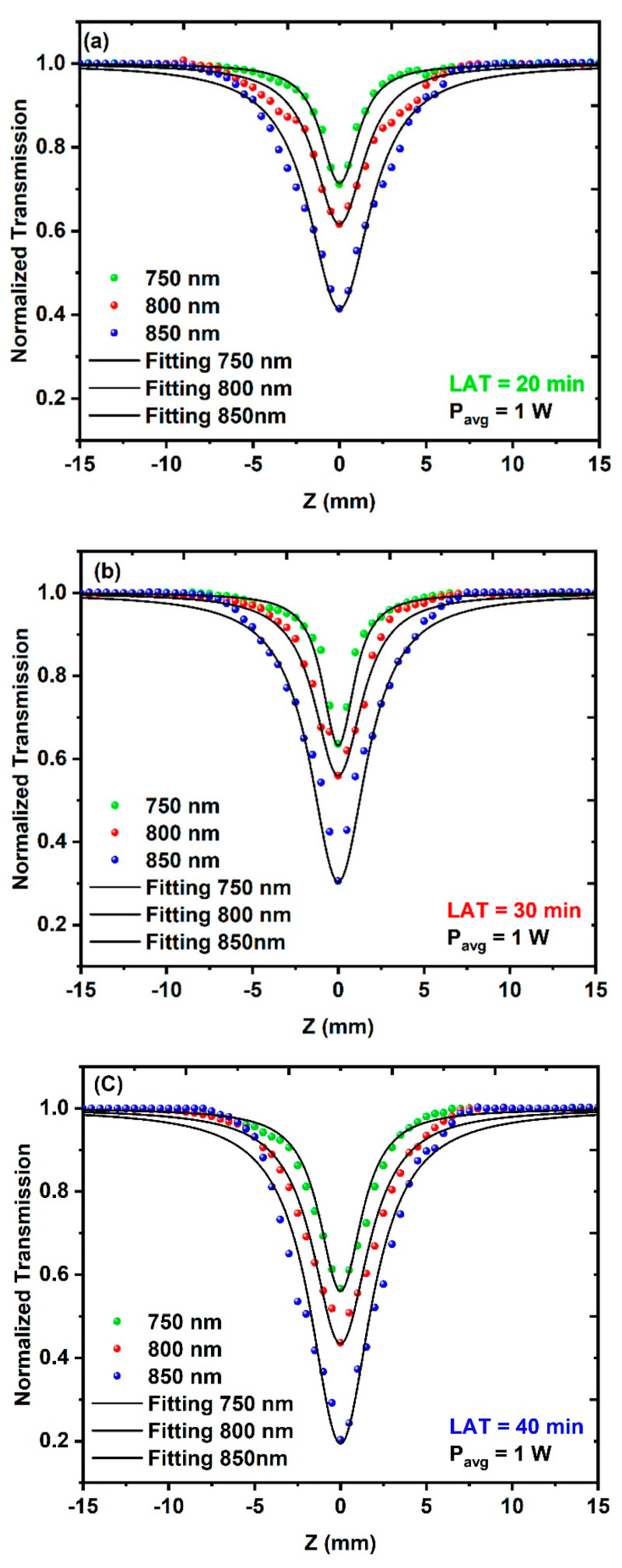
Open-aperture (OA) Z-scan measurements for different CuO NPs samples at a constant average power of 1 W: (**a**) *λ* = 750 nm, (**b**) *λ* = 800 nm and (**c**) *λ* = 850 nm.

**Figure 11 nanomaterials-14-01674-f011:**
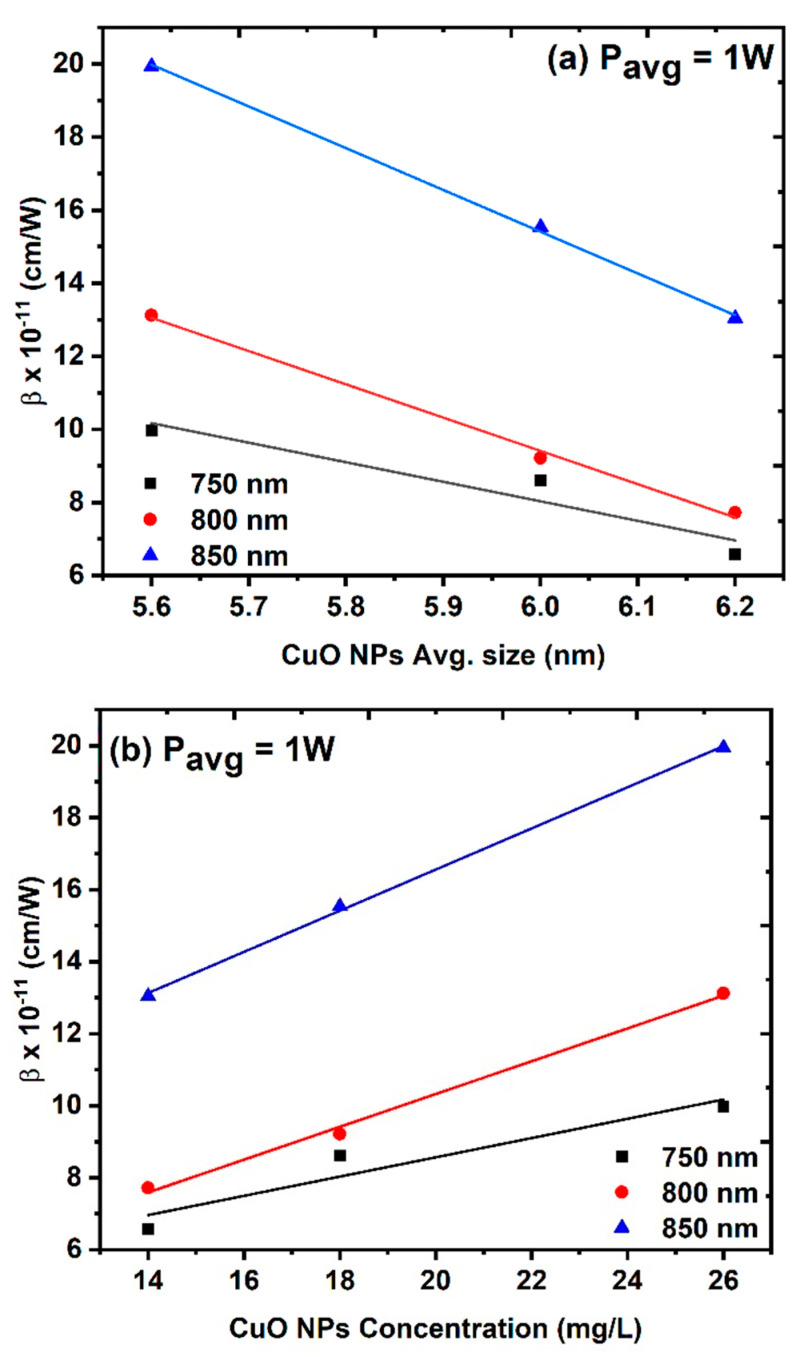
The effect of the excitation wavelength on the NLA coefficient at a constant average power of 1 W: (**a**) NLA coefficient as a function of the average size of CuO NPs and (**b**) NLA coefficient as a function of the CuO NPs concentration.

**Figure 12 nanomaterials-14-01674-f012:**
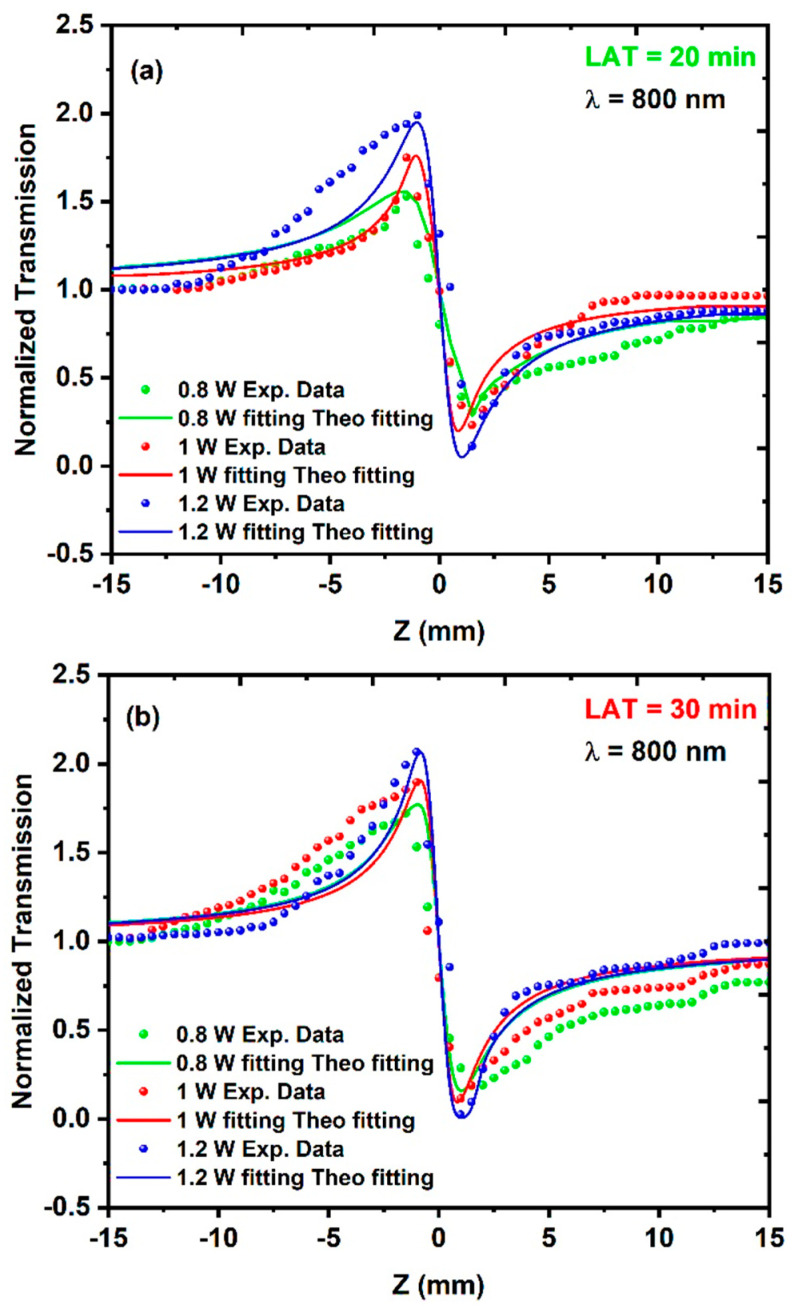
Closed-aperture (CA) Z-scan measurements for different ablation times of the CuO NPs samples at a constant excitation wavelength of 800 nm: (**a**) LAT = 20 min, (**b**) LAT = 30 min and (**c**) LAT = 40 min.

**Figure 13 nanomaterials-14-01674-f013:**
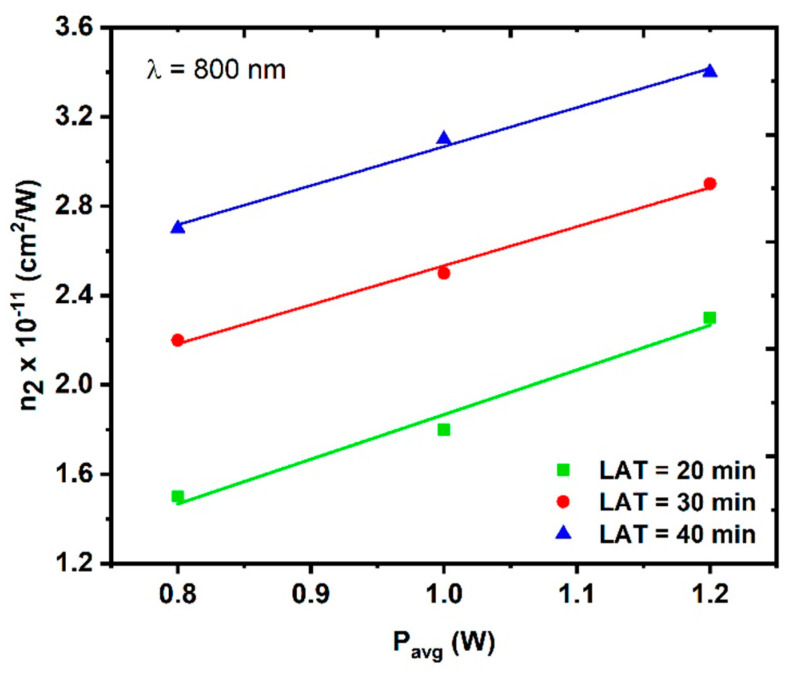
Variations in the nonlinear refractive indices of the CuO NPs samples at a constant excitation wavelength of 800 at LAT = 20 min, LAT = 30 min and LAT = 40 min.

**Figure 14 nanomaterials-14-01674-f014:**
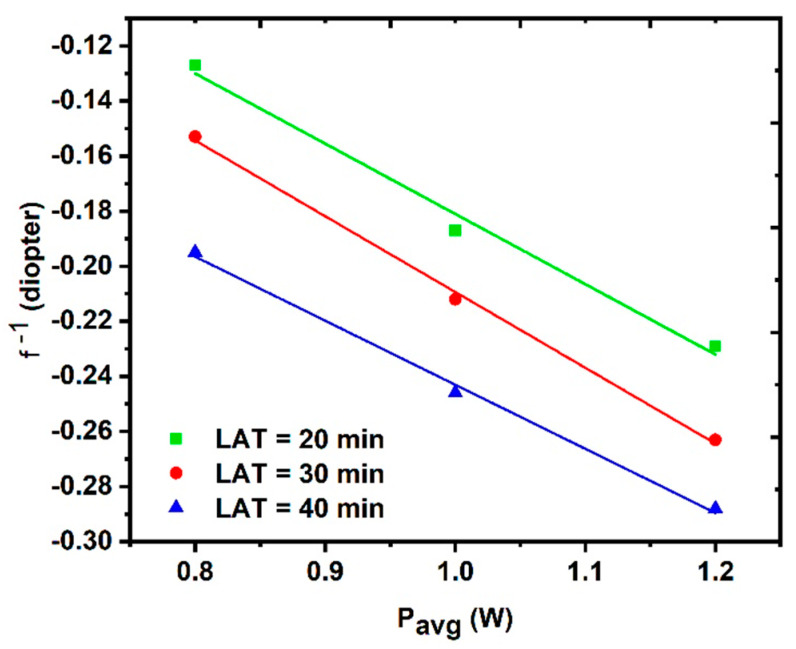
The dioptric power of the Kerr lens as a function of different average powers for LAT = 20 min, LAT = 30 min, and LAT = 40 min.

**Figure 15 nanomaterials-14-01674-f015:**
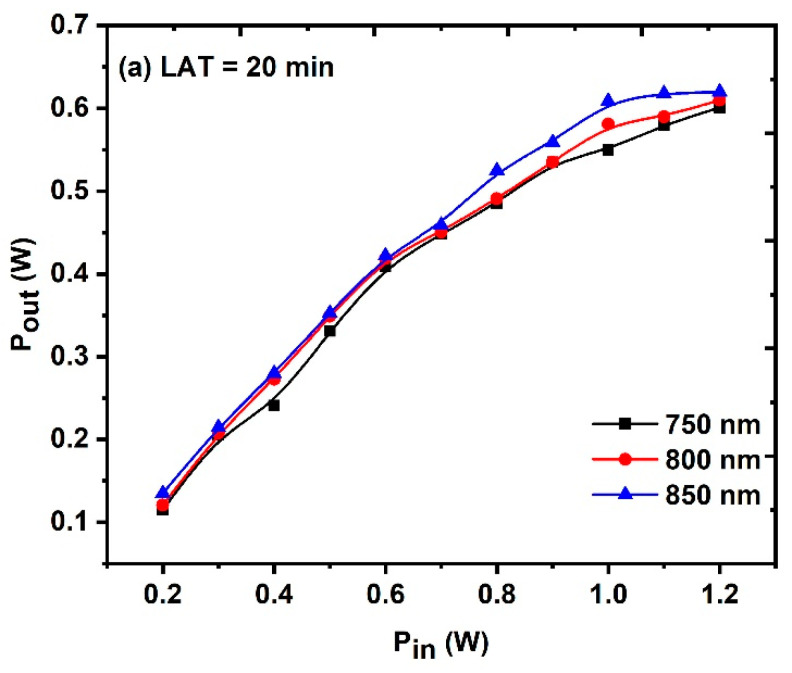
Optical limiter behavior of the CuO NPs sample (**a**) LAT = 20 min, (**b**) LAT = 30 min, and (**c**) LAT = 40 min.

**Table 1 nanomaterials-14-01674-t001:** Three-photon absorption cross-sections for the 14 mg/L and 18 mg/L and 26 mg/L CuO NPs concentrations.

*P_avg_* (W) at *λ* = 800 nm	$2PACS\;\times \;10^{-55}\;\left({\mathrm{cm}}^{4}\;s/\mathrm{Photons}\right)$		
	14 mg/L	18 mg/L	26 mg/L
0.8	1.38	1.25	1.15
1	2.27	2.11	2.07
1.2	4.42	4.01	3.06

## Data Availability

The data that underlie the results that are presented in this paper are not publicly available at this time but can be obtained from the authors upon reasonable request.

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
