# Peer review of "Using Femtosecond Laser Light to Investigate the Concentration- and Size-Dependent Nonlinear Optical Properties of Laser-Ablated CuO Quantum Dots"

_nanomaterials, 2024, doi:10.3390/nano14201674_

Round 1

Reviewer 1 Report

Comments and Suggestions for Authors

The paper “Using femtosecond laser light to explore the size and concentration-dependent nonlinear optical properties of copper oxide quantum dots created in distilled water by laser ablation” is interesting and solid work. Nevertheless, the paper would gain significantly by reorganizing the manuscript. In any case, the title is very long and could be shortened somewhat. In the introduction, various works on generating and characterizing nanoparticles are cited. On the one hand, I think the introduction could be tightened up a bit, and on the other hand, some additional literature references could be given. Several papers deal with the generation and characterization of nanoparticles, e.g., the paper by Zhang et al. (https://doi.org/10.1021/acs.chemrev.6b00468) or Kazakevich et al. (https://doi.org/10.1016/j.apsusc.2005.06.059). In the introduction, the research question should be specified in more detail, i.e., in which areas the present work goes beyond the current state of knowledge.

The devices are briefly presented in the next section, but in section 2.2, the results are also presented. I think it would make more sense to separate the description of the devices, samples, and methods used from the results. The devices and methods used are currently presented relatively unclear and scattered over the text, which should be changed. The description in Fig. 2 should be corrected. How were the size distributions of the NPs determined?

The following section presents the results of the investigations of the optical properties. Here, descriptions of devices and methods should be moved to a separate section, and only the results should be presented. The results of the measurements are presented in various figures. The information content, for example, of the different z-scan measurements in Figs. 8 and 10 are unclear, primarily as the results are hardly described. It is possible that Figs. 9 and 11 would suffice here. This also applies to Fig. 12. In Fig. 15, the effectiveness of the NP as an optical limiter is shown as being dependent on its concentration. This should also be shown as such; the laser power is plotted in Fig. 15a-c. The correlation should be explained or displayed differently. In addition, it is again questionable whether all three figures are necessary or what information results from the representation of the different processing times. In the last paragraph, the error limits of the work are mentioned, and a limit of 10 % is specified. What is this statement based on, and how was this error rate determined?

In the Conclusions section, the results should be available with concrete statements. Here, for example, it is stated that “...decrease in the NPs size was observed as a function of the laser ablation time, which has a great impact on both NPs concentration and the energy band gap...”. This should be explicitly stated if the concentration increases or decreases. It is also a very general statement that different coefficients depend on the average power and wavelength. How do they rely on these parameters? Finally, it is stated that the NP as an optical limiter relies on the size of the NP. Section 4 states that the function depends on the concentration. Even if the two quantities are related, the results section should present and explain this statement.

Author Response

A point-by-point response to Reviewer #1 comments

Dear, 

We appreciate your excellent remarks on our manuscript, as well as your comments, corrections, and valuable ideas. We believe the following response addresses all of the reviewers' issues. The detailed revisions are listed below where we present the comments of the reviewer in italic red letters, followed by our response in blue letters.

  • The title is very long and could be shortened somewhat.

We appreciate the reviewer's valuable comments which have been taken into account in the revised version of the manuscript.

The original title has been shortened in the revised version of the manuscript to "Using femtosecond laser light to investigate the concentration- and size-dependent nonlinear optical properties of laser-ablated CuO quantum dots"

  • In the introduction, various works on generating and characterizing nanoparticles are cited. On the one hand, I think the introduction could be tightened up a bit, and on the other hand, some additional literature references could be given. Several papers deal with the generation and characterization of nanoparticles, e.g., the paper by Zhang et al. (https://doi.org/10.1021/acs.chemrev.6b00468) or Kazakevich et al. (https://doi.org/10.1016/j.apsusc.2005.06.059).

Thank you for the proposed references which are very useful and valuable. We added  "https://doi.org/10.1021/acs.chemrev.6b00468" to the references list  (# 13) and discussed it in the introduction section of the revised manuscript.

  • In the introduction, the research question should be specified in more detail, i.e., in which areas the present work goes beyond the current state of knowledge.

We appreciate the reviewer's valuable comments which have been taken into account in the revised version of the manuscript.

  • The devices are briefly presented in the next section, but in section 2.2, the results are also presented. I think it would make more sense to separate the description of the devices, samples, and methods used from the results. The devices and methods used are currently presented relatively unclear and scattered over the text, which should be changed.

We appreciate the reviewer's valuable comments which have been taken into account in the revised version of the manuscript.  Instead of drafting the manuscript from the conventional viewpoint, the authors of the current study chose to draft it from an alternative perspective.   This manuscript is divided into three sections. The first describes the process of manufacturing nanomaterials and the findings obtained from the created materials. The second portion discusses how to assess linear and nonlinear optical properties using the Z-scan approach, as well as the results obtained. Finally, in the final section, the applications of the produced nanomaterials are explored and investigated.

  • The description in Fig. 2 should be corrected. How were the size distributions of the NPs determined?

We appreciate the reviewer's valuable comments which have been taken into account in the revised version of the manuscript and highlighted in yellow in section 2.2.

We want to emphasize that TEM measurements confirmed the morphology and size of the CuO nanoparticles within the distillate water. The TEM images of colloidal CuO nanoparticles formed by laser light are shown in the insets of Fig. 2 a-c, and the micrographs show that the CuO nanoparticles are spherical-like. The micrographs were analyzed using ImageJ software to determine the average particle size. Figure 2a–c, illustrate the histograms of the size distribution of CuO nanoparticles at a constant average power of 600 mW and different ablation times of 20 min, 30 min, and 40 min. Gaussian fits of the histogram provided average nanoparticle sizes of 6.2 nm, 6 nm, and 5.6 nm, respectively.

  • The results of the measurements are presented in various figures. The information content, for example, of the different z-scan measurements in Figs. 8 and 10 are unclear, primarily as the results are hardly described. It is possible that Figs. 9 and 11 would suffice here. This also applies to Fig. 12. In Fig. 15, the effectiveness of the NP as an optical limiter is shown as being dependent on its concentration. This should also be shown as such; the laser power is plotted in Fig. 15a-c. The correlation should be explained or displayed differently.

We appreciate the reviewer's helpful remarks which have been taken into account in the revised version of the manuscript and highlighted in yellow.

For further clarity, Fig. 8  shows the experimental results of the open aperture (OA) measurements for the three different CuO NPs samples. During these measurements, the samples were irradiated to different average powers of 0.8 W, 1 W, and 1.2 W at a constant excitation wavelength of 800 nm. The nonlinear absorption coefficient was estimated from the experimental data in Fig. 8 and displayed in Fig. 9. as a function of CuO NP size and concentration. The effect of excitation wavelength on the nonlinear absorption coefficient was measured and shown in Figs. 10 and 11. Similar measurements were made in closed aperture (CA) to determine the dependence of the nonlinear refractive index on the excitation wavelength and average power (Figs. 12, 13, and 14).

  • In the last paragraph, the error limits of the work are mentioned, and a limit of 10 % is specified. What is this statement based on, and how was this error rate determined?

We appreciate the reviewer comments, which we have now considered in the revised version of the manuscript and highlighted in yellow.

The experimental error in the obtained NLO parameters was about 10%, which mainly originated from the determination of the irradiance distribution used in the experiment, i.e., beam waist, pulse width, and laser power calibration.

  • In the Conclusions section, the results should be available with concrete statements. Here, for example, it is stated that “...decrease in the NPs size was observed as a function of the laser ablation time, which has a great impact on both NPs concentration and the energy band gap...”. This should be explicitly stated if the concentration increases or decreases. It is also a very general statement that different coefficients depend on the average power and wavelength. How do they rely on these parameters?

We appreciate the reviewer's feedback. The conclusions section of the revised version of the manuscript has been revised to take into account the reviewer's comments.

  • Finally, it is stated that the NP as an optical limiter relies on the size of the NP. Section 4 states that the function depends on the concentration. Even if the two quantities are related, the results section should present and explain this statement.

We appreciate the reviewer comments, which we have now considered in the revised version of the manuscript and highlighted in yellow.

Reviewer 2 Report

Comments and Suggestions for Authors

First, authors should re-write significant parts of the text and decrease plagiarism.
The second author used nanosecond lasers to create materials. In this case, melting and droplet hydrodynamics are dominant processes that determine the final structure. The ablation is the dominant process in the case of the interaction at femtoscale with matter  (when laser interaction is faster than melting)-this fact is not discussed
Figure 4 shows fit with only 3 points, not enough for conclusion
The authors talk about quantum dots, but I did not see any emission spectra of their quantum dots in the text. We cannot talk about quantum dots without measurable emission (size dependent/quantum confinement).

A lot of graphs have only 3 points not enough for a good fit.

Based on all the criticisms mentioned above, I recommend rejecting this draft in the presented form.

Author Response

A point-by-point response to Reviewer #2 comments

Dear, 

We appreciate your excellent remarks on our manuscript, as well as your comments, corrections, and valuable ideas. We believe the following response addresses all of the reviewers' issues. The detailed revisions are listed below where we present the comments of the reviewer in italic red letters, followed by our response in blue letters.

  • First, authors should re-write significant parts of the text and decrease plagiarism.

We appreciate the reviewer's valuable comments which have been taken into account in the revised version of the manuscript. The manuscript has been modified to reduce plagiarism and improve the language.

  • The second author used nanosecond lasers to create materials. In this case, melting and droplet hydrodynamics are dominant processes that determine the final structure. The ablation is the dominant process in the case of the interaction at femtoscale with matter (when laser interaction is faster than melting)-this fact is not discussed.

We'd like to thank the reviewer for bringing out such an intriguing and significant subject. The reviewer's proposed process is briefly mentioned in the revised manuscript from the beginning on line 81 (highlighted in yellow) without going through the mathematical model, as the present manuscript's goal was focused on the use of nanoparticles produced by the nanosecond laser ablation process to explore the nonlinear optical properties.  In the future, we intend to investigate the mathematical model given by the reviewer in all of the ongoing laser ablation experiments.

  • Figure 4 shows fit with only 3 points, not enough for conclusion.

We sincerely appreciate all of reviewer valuable comments and suggestions. The measurements were done for different ablation times ranging from 10 minutes to 60 minutes with a 10-minute step. The measured molar concentration of CuO NPs behaves linearly as a function of the ablation time. Three CuO NP samples were selected at 20 minutes, 40 minutes, and 60 minutes to explore the nonlinear optical properties, which is why Fig. 4 only shows three points.

  • The authors talk about quantum dots, but I did not see any emission spectra of their quantum dots in the text. We cannot talk about quantum dots without measurable emission (size dependent/quantum confinement).

We appreciate the reviewer's valuable thoughts. The TEM results showed that all the CuO NPs sample sizes were below 10 nm, which suggests that we are in the region of quantum dots according to the generally accepted definition.  Furthermore, as we previously stated, the purpose of this manuscript was to investigate the nonlinear optical properties of the ablated CuO NPs. In the future, we intend to investigate size-dependent/quantum confinement, which we believe is a good topic to study.

Reviewer 3 Report

Comments and Suggestions for Authors

In the article entitled “Using femtosecond laser light to explore the size and concentration-dependent nonlinear optical properties of copper oxide quantum dots created in distilled water by laser ablation” show how the nonlinear absorption properties and nonlinear refraction of CuO nanoparticles vary as a function of their size, concentration and experimental conditions, such as wavelength and excitation power. They use Z-scan, and from this saturable inverse absorption behaviors were identified and it is observed that the nanoparticles act as an effective optical limiter, which makes them useful for optical shielding applications.

It is concluded that the laser ablation time influences the size reduction of the nanoparticles, which in turn affects the concentration and energy bandgap of the CuO NPs, allowing their use in various nonlinear optical applications.

However, I think that some improvements should be considered by the authors.

1.      The discussion of the results obtained on saturable inverse absorption and optical limiting behavior could be extended. What is the impact of these results in real applications (such as protection of optical sensors). Compare the results with previous studies in more detail.

2.      Although it is mentioned that pulsed laser ablation is sufficient, there is no comparison with other nanoparticle synthesis techniques. Include a comparative analysis with other synthesis methods (sol-gel, precipitation to mention a few).

3.      The study focuses on a limited range of absorption times, has a wider range been studied? Testing with different ablation strengths, wavelengths or synthesis media could provide a more complete understanding of the behavior of copper oxide nanoparticles.

4.      Include a more in-depth review on recent work on NLO in CuO nanoparticles.

5.      In addition to their potential application as optical limiters, do they have applications in telecommunications, eye protection or signal processing?

6.      The description of the experimental methods could be more detailed on the experimental controls, such as the controls for the experiment, such as techniques to ensure homogeneity of the samples.

Comments on the Quality of English Language

Minor editng of English language requiered

Author Response

A point-by-point response to Reviewer #3 comments

Dear, 

We appreciate your excellent remarks on our manuscript, as well as your comments, corrections, and valuable ideas. We believe the following response addresses all of the reviewers' issues. The detailed revisions are listed below where we present the comments of the reviewer in italic red letters, followed by our response in blue letters.

  • The discussion of the results obtained on saturable inverse absorption and optical limiting behavior could be extended. What is the impact of these results in real applications (such as protection of optical sensors). Compare the results with previous studies in more detail.

We appreciate the reviewer's helpful remarks, which have been included and marked in yellow in the results section of the revised version of the manuscript.

  • Although it is mentioned that pulsed laser ablation is sufficient, there is no comparison with other nanoparticle synthesis techniques. Include a comparative analysis with other synthesis methods (sol-gel, precipitation to mention a few).

We value the reviewer's insightful remarks and recommendations. A brief discussion of the various methods for producing NPs has been added in the revised version of the manuscript as follows:

“There are three basic techniques for NPs synthesis: physical, chemical, and biological. The physical technique is also known as the top-down approach, whereas the chemical and biological approaches are commonly referred to as the bottom-up approach. The biological approach is also known as green systems of NPs [19]. Top-down techniques separate bulk materials into nanostructured materials. Top-down techniques include mechanical milling, laser ablation, etching, sputtering, and electro-explosion [20]. The bottom-up method, also known as the constructive method, includes the construction of materials ranging from atoms to clusters to nanoparticles. Bottom-up processes include chemical vapor deposition, sol-gel, spinning, pyrolysis, and biological synthesis [21].”

[19] Baig, Nadeem and Kammakakam, Irshad and Falath, Wail (2021). Nanomaterials: a review of synthesis methods, properties recent progress, and challenges. Mater. Adv., volume  2,

 issue  6, pages 1821-1871.

[20] Bawoke Mekuye and Birhanu Abera (2023) .Nanomaterials: An overview of synthesis, classification, characterization, and applications. Nano Select,  Volume 4, Issue 8, Pages 486-501

[21] T. C. Mokhena, M. J. John, M. A. Sibeko, V. C. Agbakoba, M. J. Mochane, A. Mtibe, T. H. Mokhothu, T. S. Motsoeneng, M. M. Phiri, M. J. Phiri, P. S. Hlangothi & T. G. Mofokeng  (2020). Nanomaterials: Types, Synthesis and Characterization. In: Srivastava, M., Srivastava, N., Mishra, P., Gupta, V. (eds) Nanomaterials in Biofuels Research. Clean Energy Production Technologies. Springer, Singapore.

  • The study focuses on a limited range of absorption times, has a wider range been studied? Testing with different ablation strengths, wavelengths or synthesis media could provide a more complete understanding of the behavior of copper oxide nanoparticles.

We sincerely appreciate all of reviewer valuable comments and suggestions. The purpose of this manuscript was to investigate the nonlinear optical properties of the ablated CuO NPs. In the future, we intend to investigate the CuO NPs size, morphology, and concentration dependent on laser ablation power, wavelength, and ablation times, which we believe is a good topic to study.

  • Include a more in-depth review on recent work on NLO in CuO nanoparticles.

We appreciate the reviewer's helpful remarks, which have been included and marked in yellow in the results section of the revised version of the manuscript.

  • In addition to their potential application as optical limiters, do they have applications in telecommunications, eye protection or signal processing?

We appreciate the reviewer's thoughtful comment. We have addressed this issue in the introduction section of the revised manuscript, as NPs do indeed offer enormous potential in optical computing and signal processing. 

  • The description of the experimental methods could be more detailed on the experimental controls, such as the controls for the experiment, such as techniques to ensure homogeneity of the samples.

We appreciate the reviewer's helpful input. More information has been added to the experimental part of the revised manuscript.

Round 2

Reviewer 2 Report

Comments and Suggestions for Authors

The authors provided detailed and correct answers to all referee questions. Based on that fact, I recommend publication of the revised draft in the present form.